# Vector dominance, one flavored baryons, and QCD domain walls from the "hidden" Wess-Zumino term

Avner Karasik

*Department of Applied Mathematics and Theoretical Physics, University of Cambridge, Cambridge, CB3 OWA, UK*

`avnerkar@gmail.com`

**Abstract**

We further explore a recent proposal that the vector mesons in QCD have a special role as Chern-Simons fields on various QCD objects such as domain walls and the one flavored baryons. We compute contributions to domain wall theories and to the baryon current coming from a generalized Wess-Zumino term including vector mesons. The conditions that lead to the expected Chern-Simons terms and the correct spectrum of baryons, coincide with the conditions for vector meson dominance. This observation provides a theoretical explanation to the phenomenological principle of vector dominance, as well as an experimental evidence for the identification of vector mesons as the Chern-Simons fields. By deriving the Chern-Simons theories directly from an action, we obtain new results about QCD domain walls. One conclusion is the existence of a first order phase transition between domain walls as a function of the quarks' masses. We also discuss applications of our results to Seiberg duality between gluons and vector mesons and provide new evidence supporting the duality.

December 8, 2020

# 1   Introduction

QCD is a theory that we understand very well at the two edges of the RG flow. At very high energies it is described as a weakly interacting $SU(N)$ gauge theory coupled to $N_f$ fundamental fermions. By assuming confinement and chiral symmetry breaking, the low energy theory is described as an $SU(N_f)$ non-linear sigma model with a level $N$ Wess-Zumino (WZ) term [1, 2]. The WZ term is fixed uniquely by anomaly matching conditions. This model and in particular the WZ term are extremely successful in combining deep theoretical ideas with concrete measurements. Once we leave the deep infrared (IR) limit and increase the energy, we lose theoretical control over the physics. In addition to higher derivatives terms, a zoo of mesons come back to life, which results in many possible interactions. Ideally, we would like to find theoretical arguments that reveal a hidden order in the theory and restrict the space of couplings. In particular, we will be interested here in the effective theory containing in addition to pions, also the $\eta'$ field and the $U(N_f)$ vector mesons known as $\rho$ and $\omega$ which we will denote collectively by $V$. The most controlled way to add the $\eta'$ to the chiral Lagrangian is by taking the large $N$ limit. When $N$ is large, the axial symmetry $U(1)_A$ becomes a good symmetry of the theory. The spontaneous breaking of $U(1)_A$ leads to an extra Nambu-Goldstone (NG) boson which is the $\eta'$ meson. Indeed, the mass of the $\eta'$ field is suppressed in the large $N$ limit, $m_{\eta'}^2 \sim \frac{1}{N}$ [3, 4]. For the vector mesons, there are two phenomenological principles that are commonly used when writing their effective theory. The first is the hidden local symmetry (HLS) principle [5, 6] which will be

reviewed in section 2, and the second is Vector Dominance (VD) [7, 8] which will be reviewed in section 3. These two principles restrict the space of couplings and increase the predictive power of the theory. Yet, a good theoretical explanation for why these principles are correct is absent. Recently [9, 10], it has been conjectured that the vector mesons have a special role as the Chern-Simons (CS) vector fields on various QCD objects, such as domain walls (DWs), interfaces, and $N_f = 1$ baryons. In this work we show that the identification of the vector mesons as the CS fields is intimately related to HLS and VD, at least in the large $N$ limit. This story can be told in two ways,

1. Phenomenology→Theory: The effective theory for vector mesons can be written using the assumptions of HLS and VD. The emergent theory on domain walls can be derived classically from this effective Lagrangian. The emergent theory contains a CS term with vector mesons as the vector fields. In addition, corrections to the baryon current can be computed. The full baryon current reproduces the correct baryonic spectrum in the sense that both skyrmions and the $N_f = 1$ baryon introduced in [11] have charge 1 under it. When telling the story in this direction, we can say that the experimental results prove the conjecture that the vector mesons are indeed the CS fields.

2. Theory→Phenomenology: Assuming that the vector mesons are the CS fields, we can demand that the low energy effective theory will reproduce the correct domain wall theories and baryonic spectrum, as expected from theory. Then VD is automatically satisfied. When telling the story in this direction, we can say that our demand provides a theoretical explanation to the 50 year old idea of VD.

The main actor in this story is the "hidden" part of the WZ action, we will denote by the hWZ action. The hWZ action contains all the additional terms that are odd under the intrinsic parity $U \to U^\dagger$, when vector mesons are included. Unlike the regular WZ term which is uniquely fixed by topology and anomalies, there is a family of consistent hWZ actions, parametrized by three real numbers.[1] When coupling the theory to (background or dynamical) gauge field for some global $U(1)$ symmetry, there is an extra improvement term such that the hWZ action is then parametrized by four real parameters. The hWZ action has been studied a lot mainly from the phenomenological point of view. See for example [12] for a comprehensive review. The focus in the existing literature is on the application of the hWZ action to processes odd under the intrinsic parity $U \to U^\dagger$. The full WZ action is the only part of the action violating the intrinsic parity, and as such, it is solely responsible for all odd processes such as $\omega \to \pi\pi\pi$, $\pi_0 \to \gamma\gamma$ where $\gamma$ is the photon field, and more. VD in this context is the observation that certain processes are dominated by an exchange of an internal vector meson. This will happen if the direct vertices don't appear explicitly in the Lagrangian.

---

[1]This is true in the large $N$ limit. At finite $N$ one can write also multi-trace operators and the freedom is larger. We will comment about it in section 7.

In particular, we will be interested in a specific type of VD where the following three vertices are absent: $\gamma\gamma\pi$, $\gamma V\pi$, and $V\pi\pi\pi$. This means for example that the decay $\pi_0 \to \gamma\gamma$ is mediated by vector mesons $\pi_0 \to VV \to \gamma\gamma$. On the same way, $\omega \to \pi\pi\pi$ is mediated by $\omega \to \rho\pi \to \pi\pi\pi$. The elimination of these three vertices gives three conditions, which leave one free parameter in the hWZ action+photon consistent with this type of VD.

In this work, we study additional applications of the hWZ action such as its contribution to the 3d effective theory on domain walls, and to the baryon current.

Baryon current: At low energies, the baryon current can be extracted from the WZ term [2]. The hWZ term gives corrections to the skyrmion current that involve the vector mesons. The correction is a total derivative and doesn't contribute to the baryon charge of any smooth configuration. The importance of these corrections comes when studying singular charged configurations such as the $N_f = 1$ baryons [10, 11]. The idea to derive the baryon current for $N_f = 1$ baryons from the hWZ action was also mentioned in [13, 14]. Very briefly, these baryons are made out of a finite $\eta' = \pi$ disc with a $U(1)_N$ CS theory living on the disc. On the ring that bounds the disc, $\eta'$ is not well defined which makes the configuration singular. As in the quantum Hall effect, the CS field gives rise to a chiral edge mode on the ring. The charge of the baryon comes from two orthogonal windings- $\eta'$ around the ring, and the edge mode along the ring. For the baryon current derived from the full WZ term to reproduce the correct baryonic spectrum, the CS fields should be identified with the $\omega$ vector meson, and one condition on the parameters of the hWZ action must be imposed. Surprisingly, this condition is satisfied if one assumes VD.

$\eta'$ Domain walls: The vacuum of massive QCD at $\theta = \pi$ breaks time reversal symmetry, which implies the existence of DWs connecting the two vacua [15]. Given the effective theory of pions and vector mesons, the emergent theory on the DW can be extracted classically. First, we can make connection with the $N_f = 1$ baryons, by requiring that on the $\eta' = \pi$ DW, indeed there will be an emergent $U(1)_N$ CS theory. This demand gives another condition on the hWZ parameters. This condition is also satisfied if one assumes VD. The two other parameters can be fixed by demanding that the emergent theory on the $\eta' = \pi$ wall will have the form of a CS action also for $N_f \geq 2$. This gives two extra conditions. All together we have four conditions based on theoretical arguments that fix the hWZ action+ external $U(1)$ gauge field completely.[2] The three conditions for VD are contained inside these four conditions.

In addition to establishing the relation between VD, DW, and Baryons, the identification of vector mesons as the CS fields on DWs has several interesting consequences:

Phase transitions of DWs: It is an open question whether the Yang-Mills (YM) DW is connected continuously to the $m \ll \Lambda$ DW in QCD or not. If the answer is positive, one could expect to be able to find a 3d theory describing the DW for any value of the

---

[2]Notice that the external $U(1)$ gauge field was the photon when discussing VD, and a background field for $U(1)_B$ when deriving the baryon current.

quarks' masses [15].[3] The other option is that there is a first order phase transition between DWs such that the YM DW is connected continuously to some metastable DW when $m \ll \Lambda$. We will argue that this is indeed the case and the YM DW is connected continuously to the $\eta' = \pi$ DW which is metastable when $m \ll \Lambda$. This can be done by studying the contribution to DW theories coming from the hWZ action. The result contradicts the proposal of [15] which is based on the assumption of no phase transition. We will also provide additional arguments supporting the phase transition scenario and give a new proposal for the DW theory.

Mesons-Gluons Duality: Another application is related to a conjectured Seiberg-like duality between gluons and vector mesons [9, 10, 17–19]. Very roughly, the idea is as follows. You start at high energies from an SU(N) gauge theory of gluons coupled to $N_f$ quarks. As the energy is lowered, the theory becomes strongly coupled until at some point a dual description of the theory appears. The dual description contains $U(N_f)$ gauge theory coupled to some matter. The $U(N_f)$ gauge fields become massive via the Higgs mechanism and are identified with the vector mesons. The question is to what extent this duality is correct. The fact that the two theories flow to the same chiral Lagrangian in the deep IR is known. Assuming the hidden local symmetry principle is correct, the two theories give rise to the same physics also above the deep IR when the vector mesons are treated as dynamical. Can we push this duality even higher along the RG flow? is there a point (presumably related to the chiral restoration point) where the Higgs vev goes to zero, and the vector mesons become true massless gauge fields? We don't know the answers to these hard questions but we can at least give some new evidence supporting this picture. The first comes from CS dualities on the DWs. Dualities of the form

$$U(N_f)_N \simeq SU(N)_{-N_f} \ ,$$ 

(1.1)

can be used to map the $U(N_f)$ vector mesons to the $SU(N)$ gluons on the domain wall. Another hint comes from our conjecture that the YM DW is connected continuously to the $\eta' = \pi$ DW at small masses. Consider $N_f = 1$ for simplicity. By deforming the mass of the quark continuously from $m \ll \Lambda$ to $m \gg \Lambda$, the $U(1)_N$ Chern-Simons-Higgs (CSH) theory should flow continuously to pure $U(1)_N$ CS theory which is the theory on the YM DW [20]. This implies that the mass of the $\omega$ vector meson on the domain wall goes to zero as one increases the mass of the quark. This is in contrast to the intuition that masses of composite particles increases as the mass of their constituents increases. Moreover, $\omega$ should survive the $m \to \infty$ limit and transform continuously into some kind of a glueball (because that's the only thing that exists in pure YM). The fact that there exists a regime in which the $\omega$ vector meson (or a continuous deformation

---

[3]There is also the possibility that the 3d theory on the DW is not well defined everywhere. This can happen if the bulk excitations are comparable or lighter than the DW excitations, and the interactions between them are not suppressed in any sense. See [16] for a general discussion on this point. We ignore this issue throughout the paper.

of it) is massless and has a dual description in terms of gluons also support the duality between vector mesons and gluons.

Cusp potential for $\eta'$: Another important part of our work is related to the formation of the cusp potential of the $\eta'$ meson in the large $N$ limit, [4, 21, 22]

$$V_{\eta'} = \frac{1}{2} F_\pi^2 M_{\eta'}^2 \min_{k \in \mathbb{Z}} (\eta' - 2\pi k)^2 + \dots \ . \tag{1.2}$$

The conventional explanation for the origin of this potential is that it comes from integrating out the gluonic topological density [22]. It has been conjectured in [10] that a dual description exists in which the cusp is generated from integrating out the vector mesons. This conjecture was motivated by the CS duality between gluons and vector mesons on the DWs. The hWZ action studied here enables us to write the effective coupling between $\eta'$ and vector mesons required to derive the mentioned CS duality. This conjecture is a key assumption in the identification of the vector mesons as the CS vector fields on the DWs. The details of how the potential is generated will be studied elsewhere [23].

The outline of the paper is as follows. In section 2 we will review the idea of HLS and introduce the hWZ action. In section 3 we will review the conditions on the hWZ parameters for the type of VD we impose. This computation already appeared in [12]. In section 4 we compute corrections to the skyrmion current coming from the hWZ action. We will find the condition that reproduces the correct baryonic spectrum and show that it agrees with VD. In section 5 we study domain walls. We start from DWs in $N_f = 1$ QCD in 5.1, then move on to $N_f \geq 2$ QCD in 5.2 and 5.3. In section 6 we review the mechanism for generating the cusp potential of $\eta'$ [22] and present the conjecture for the dual description using vector mesons. We finish with a summary of the conditions on the hWZ parameters and some comments about finite $N$ in section 7. Some technical computations appear in the appendix A.

# 2 Generalized WZ term from hidden local symmetry

We will begin this section by presenting the hWZ action. Recall that the WZ term can be written as [2]

$$S_{WZ,U} = -\frac{iN}{240\pi^2} \int_{\mathcal{B}_5} \Gamma_U \ , \tag{2.1}$$

where

$$\Gamma_U = dUU^\dagger dUU^\dagger dUU^\dagger dUU^\dagger dUU^\dagger \ . \tag{2.2}$$

Here and later, there is an implicit trace in flavor space, and all the forms are assumed to be contracted with the $\wedge$ product. The integration is over a 5 dimensional manifold $\mathcal{B}_5$ whose boundary is the 4 dimensional world $\mathcal{M}_4 = \partial\mathcal{B}_5$. Miraculously, the theory

doesn't depend on the fifth dimension for every $N \in \mathbb{Z}$ in (2.1), thanks to

$$-\frac{in}{240\pi^2} \int_{\mathcal{M}_5} \Gamma_U = 2\pi\mathbb{Z} \; \forall \; n \in \mathbb{Z} \; , \tag{2.3}$$

for every closed manifold $\mathcal{M}_5$. The integer is fixed to be the number of colors $N$ by anomaly matching conditions. While (2.1) is uniquely fixed at low energies, we want to study a more fundamental theory and include in addition to pions, also the vector mesons. Any consistent action that reduces to (2.1) when integrating out the vector mesons, is a possible "orange" completion (as opposed to uv completion here we are just a little bit above the infrared). We will introduce the vector mesons into the chiral Lagrangian using the idea of hidden local symmetry and classify the space of allowed completions for the WZ term. In the next sections, we will use various theoretical arguments to fix the WZ term completely.

In the hidden local symmetry principle, we write $U$ as a product of two unitary matrices

$$U = \xi_L^\dagger \xi_R \; , \tag{2.4}$$

where the redundant transformations $\xi_{R,L} \to h\xi_{R,L}$ with $h \in SU(N_f)$ are coupled to dynamical gauge fields $V$ which transform as $V \to hVh^\dagger + ihdh^\dagger$. In addition, the global $SU(N_f)_L \times SU(N_f)_R$ symmetries act as

$$\xi_R \to \xi_R g_R^\dagger \; , \; \xi_L \to \xi_L g_L^\dagger \; . \tag{2.5}$$

We also introduce the covariant derivative and the field strength

$$D\xi_I \xi_I^\dagger = d\xi_I \xi_I^\dagger - iV \; , \; F = dV - iV^2 \; . \tag{2.6}$$

A convenient shorthand notation we will use throughout the paper is

$$R = d\xi_R \xi_R^\dagger \; , \; L = d\xi_L \xi_L^\dagger \; , \; R_D = D\xi_R \xi_R^\dagger \; , \; L_D = D\xi_L \xi_L^\dagger \; . \tag{2.7}$$

The most general two derivatives Lagrangian consistent with the above symmetries is

$$\mathcal{L} = \frac{F_\pi^2}{4} \text{tr} \left( \partial_\mu (\xi_R^\dagger \xi_L) \partial^\mu (\xi_L^\dagger \xi_R) \right) - \frac{aF_\pi^2}{4} \text{tr} \left[ D_\mu \xi_L \xi_L^\dagger + D_\mu \xi_R \xi_R^\dagger \right]^2 - \frac{1}{4g^2} F_{\mu\nu}^2 \; , \tag{2.8}$$

where $a$ is some dimensionless free parameter and $g$ is the coupling constant.

In the unitary gauge $\xi_R = \xi_L^\dagger = \xi$, this is written as

$$\mathcal{L} = \frac{F_\pi^2}{4} \text{tr} \left( \partial_\mu U^\dagger \partial^\mu U \right) - \frac{aF_\pi^2}{4} \text{tr} \left[ \partial_\mu \xi \xi^\dagger + \partial_\mu \xi^\dagger \xi - 2iV_\mu \right]^2 - \frac{1}{4g^2} F_{\mu\nu}^2 \; . \tag{2.9}$$

In addition to the usual kinetic terms for the pions and the vector fields, the Lagrangian contains a mass term for the vector fields with $m_V^2 \sim ag^2 F_\pi^2$, and interactions between the vectors and the pions. Now we will present the most general hWZ action in the

theory. By hWZ action we mean all the terms whose Lorentz indices are contracted using the $\epsilon$ tensor, similar to (2.1). In addition, we demand that the action is gauge invariant under the hidden gauge transformations, consistent with the global symmetries (2.5) and with time reversal symmetry that acts on the fields as

$$U \leftrightarrow U^\dagger , \ \xi_L \leftrightarrow \xi_R , \ V \to V . \tag{2.10}$$

We will also simplify the action by taking the large $N$ limit in which only single trace (in flavor space) operators are considered. The most general hWZ Lagrangian that can be added to (2.1) is [12]

$$\mathcal{L}_{hWZ} = \frac{N}{16\pi^2} \sum_{i=1}^{3} c_i \mathcal{L}_i , \tag{2.11}$$
$$\mathcal{L}_1 = i(L_D R_D^3 - R_D L_D^3) , \ \mathcal{L}_2 = iL_D R_D L_D R_D , \ \mathcal{L}_3 = F(R_D L_D - L_D R_D) ,$$

with $c_i \in \mathbb{R}$. It is straight forward to verify that in the deep IR, upon integrating out the vector mesons by replacing $V \to \frac{1}{2i}(R + L)$, $\mathcal{L}_{hWZ} \to 0$ and we are left only with (2.1) as expected. Explicitly, we can write (2.11) as

$$\begin{aligned}
\mathcal{L}_1 =& iLR^3 - iRL^3 + V(R^3 - L^3 + L^2R - R^2L - LRL + RL^2 + RLR - LR^2) \\
& - 2iV^2(LR - RL) - iVRVR + iVLVL - 2V^3(R - L) , \\
\mathcal{L}_2 =& iLRLR + 2V(RLR - LRL) - 2iV^2(LR - RL) - iVRVR + iVLVL - 2V^3(R - L) , \\
\mathcal{L}_3 =& (dV - iV^2)(RL - LR) + i(dVV + VdV)(R - L) + 2V^3(R - L) .
\end{aligned} \tag{2.12}$$

In this prescription, there is a family of consistent hWZ actions parameterized by three real numbers $\{c_i\}$. In addition, we can couple the theory to a $U(1)$ (dynamical or background) gauge field for some global $U(1)$,

$$\xi_{R,L} \to \xi_{R,L} e^{-iQ\alpha} , \ A \to A - d\alpha , \tag{2.13}$$

Here $Q$ is the diagonal matrix of charges, and $A$ is the gauge field. Two important cases we will discuss are when $A$ is the photon (see section 3), and when $A$ is a background $U(1)_B$ field (see section 4). When $A$ is included, we need to redefine the covariant derivative accordingly,

$$R_A = R_D - iA\xi_R Q\xi_R^\dagger , \ L_A = L_D - iA\xi_L Q\xi_L^\dagger . \tag{2.14}$$

The WZ action is modified due to this gauging in several ways. First, all the covariant derivatives in (2.11) should be replaced with $R_D, L_D \to R_A, L_A$. Second, there are two (not gauge invariant) four dimensional terms that should be added to (2.1) in order to maintain gauge invariance as was shown in [2]. Notice that the derivatives in (2.1) are not replaced by covariant derivatives in this formalism. Third, there is a freedom to add to the hWZ action, the gauge invariant 4d term

$$-\frac{Nc_4}{32\pi^2} dAQ[\xi_R^\dagger(R_A L_A - L_A R_A)\xi_R + \xi_L^\dagger(R_A L_A - L_A R_A)\xi_L] , \tag{2.15}$$

with $c_4 \in \mathbb{R}$.

Understanding which completion is the correct one is important both from the theoretical and phenomenological point of view. Our proposal for the hWZ action is

$$c_1 = \frac{2}{3} \ , \ c_2 = -\frac{1}{3} \ , \ c_3 = 1 \ , \ c_4 = 1 \ . \tag{2.16}$$

In the next sections we will discuss some of the applications of the hWZ action and motivate the choice of (2.16).

# 3  Intrinsic parity and vector dominance

One of the most important features of the WZ term is that this is the only term that breaks the intrinsic parity $U \to U^\dagger$. As a result, various odd processes in QCD are fixed by the WZ term. The most famous are the scattering of two kaons to three pions $K^+ K^- \to \pi^+ \pi^- \pi^0$, the decay of $\pi^0$ to two photons $\pi^0 \to \gamma\gamma$, and the 4-vertex involving $\gamma\pi^+\pi^-\pi^0$. These three don't involve vector mesons as one of the external particles, and the leading contribution indeed comes from $\Gamma_U$ coupled to the photon.[2] Other processes that contain vector mesons are for example $\omega \to \pi^+\pi^-\pi^0$ and $\omega \to \gamma\pi^0$.

Vector dominance (VD) [7] is a related phenomenological principle that states that when vector mesons are included, some vertices don't appear explicitly in the Lagrangian and the contribution to them is dominated by an exchange of internal vector meson. The study of VD from the hWZ action was considered a lot in the literature, see for example [8, 12]. In this section we will show that (2.16) implies VD for the vertices $V\pi\pi$, $AA\pi$ and $AV\pi$, where $A$ in this section plays the role of the photon. We are interested in studying the effective vertices obtained from expanding $U$ around the identity,

$$U = 1 + \frac{2i\Pi}{F_\pi} + ... \ , \ R = -L = \frac{id\Pi}{F_\pi} + ... \ . \tag{3.1}$$

As was shown in [2], expanding the gauged version of (2.1) results in

$$\frac{2N}{15\pi^2 F_\pi^5} \Pi d\Pi d\Pi d\Pi d\Pi + \frac{iN}{3\pi^2 F_\pi^3} AQ d\Pi d\Pi d\Pi - \frac{N}{4\pi^2 F_\pi} AdAQ^2 d\Pi \ . \tag{3.2}$$

Together with (2.11) and (2.15) we get

$$\frac{N(8 - 15c_1 + 15c_2)}{60\pi^2 F_\pi^5} \Pi (d\Pi)^4 + \frac{iN}{4\pi^2 F_\pi^3}(c_2 - c_1 + c_3)V(d\Pi)^3 - \frac{iN}{4\pi^2 F_\pi}(c_1 + c_2 - c_3)V^3 d\Pi$$

$$- \frac{N}{8\pi^2 F_\pi}c_3(dVV + VdV)d\Pi + \frac{iN(4 - 3c_1 + 3c_2 - 3c_4)}{12\pi^2 F_\pi^3}AQ(d\Pi)^3 - \frac{N(1 - c_4)}{4\pi^2 F_\pi}AdAQ^2 d\Pi$$

$$- \frac{iN(2c_1 + 2c_2 - c_3)}{8\pi^2 F_\pi}AQ(V^2 d\Pi + d\Pi V^2) + \frac{iN(c_1 + c_2)}{4\pi^2 F_\pi}AQV d\Pi V - \frac{N(c_3 - c_4)}{8\pi^2 F_\pi}AQ(d\Pi dV + d\Pi dV) \ . \tag{3.3}$$

The vertices $AA\Pi$, $AV\Pi$, $V\Pi\Pi\Pi\Pi$ vanish for $c_4 = 1$, $c_3 = c_4$, $c_1 - c_2 = c_3$ respectively. The three conditions together fix three out of the four free parameters,

$$c_1 - c_2 = c_3 = c_4 = 1 \ . \tag{3.4}$$

Notice that (2.16) is consistent with these three demands with the specific choice of $c_1 = \frac{2}{3}$, $c_2 = -\frac{1}{3}$. This type of VD means for example that the $\pi_0 \to \gamma\gamma$ decay is mediated by vector mesons: $\pi_0 \to VV \to \gamma\gamma$. Another example is $\omega \to \pi\pi\pi$. Since there is no direct vertex, the process is dominated by $\omega \to \rho\pi \to \pi\pi\pi$. This type of VD is consistent with the experimental results for $\pi^0 \to \gamma\gamma$, $\omega \to \pi_0\gamma$ and $\omega \to \pi^+\pi^-\pi^0$. See section (3.8) of [12] for the detailed computation.

# 4 Derivation of the baryon current

Another application of the WZ action is the derivation of the skyrmion current as the low energy description of the baryon current. We are going to follow the same procedure and apply it on our hWZ action. The prescription of [2] to derive the baryon current is as follows:

1. Compute the Noether current for a general vector-like $U(1)$ symmetry that acts as

$$U \to e^{iQ\alpha} U e^{-iQ\alpha} \ . \tag{4.1}$$

   This can be done by coupling the symmetry to a gauge field $A$ and reading the current from the term $-A_\mu J^\mu$ in the Lagrangian.

2. After deriving $J^\mu$, plug in $Q = \frac{1}{N}$. $U$ is invariant under this transformation $U \to e^{i\alpha/N} U e^{-i\alpha/N} = U$. Therefore, most of the contributions to $J^\mu$ will vanish.

3. The only exception is the contribution coming from the 5d WZ term. This is due to some extra integration by parts when going from 5d to its 4d boundary which changes the relative sign between two terms.

4. As a result, the baryon current is given by the skyrmion current

$$S^\mu = \frac{1}{24\pi^2} \epsilon^{\mu\nu\rho\sigma} tr(\partial_\nu U U^\dagger \partial_\rho U U^\dagger \partial_\sigma U U^\dagger) \ . \tag{4.2}$$

One might be suspicious about the degree of rigorousness of this derivation, since $U$ is not charged under $U(1)_B$ and this "limit" $Q \to \frac{1}{N}$ is ambiguous. However, at least in the large $N$ limit we can justify this derivation. The reason is that in the large $N$ limit, $U \in U(N_f)$ and we can take $tr(Q) \neq 0$. In this case, we can really approach $Q \to \frac{1}{N}$ continuously, and get the skyrmion current no matter how the limit is taken.

   We can repeat this procedure for the hWZ action by computing the current associated with the transformation

$$\xi_{R,L} \to \xi_{R,L} e^{-iQ\alpha} \implies U = \xi_L^\dagger \xi_R \to e^{iQ\alpha} U e^{-iQ\alpha} \ , \tag{4.3}$$

and take $Q = \frac{1}{N}$ in the end. We choose to accompany this transformation with the gauge transformation

$$\xi_{R,L} \to e^{\frac{i}{N}\alpha}\xi_{R,L} , \tag{4.4}$$

such that $\xi_{R,L}$ are themselves invariant. The derivation can be evaluated as follows. We can start from the full WZ action. In order to extract the baryon current we simply need to take $Q = \frac{1}{N}$ and also modify $V \to V - \frac{1}{N}A$. We already know that (2.1) gives rise to the skyrmion current (4.2). We will compute the contribution from the hWZ action (2.11) including the improvement term (2.15). The terms proportional to $c_1$ and $c_2$ in (2.11) don't contribute to the baryon current because $A$ doesn't appear in the covariant derivatives $R_D$, $L_D$. We do get contribution from $c_3$ due to the shift $V \to V - \frac{1}{N}A$. In addition, the improvement term contains $A$ explicitly. Together, we have

$$
\begin{aligned}
&- \frac{c_3 + c_4}{16\pi^2} Ad(R_D L_D - L_D R_D) \\
&= -\frac{c_3 + c_4}{8\pi^2} A(R^2 L - RL^2 + idV(R - L) - iV(R^2 - L^2)) ,
\end{aligned}
\tag{4.5}
$$

such that the current is

$$B = \frac{1}{24\pi^2}(dUU^\dagger)^3 + \frac{c_3 + c_4}{8\pi^2}(R^2 L - RL^2 + idV(R - L) - iV(R^2 - L^2)) . \tag{4.6}$$

Using $(dUU^\dagger)^3 = (R - L)^3$ we can write,

$$B = S + (c_3 + c_4)(H - S) , \tag{4.7}$$

where

$$H = \frac{1}{24\pi^2} \left[ R_D^3 - L_D^3 + 3iF(R_D - L_D) \right] , \tag{4.8}$$

is the hidden baryon current defined in [10].

In the $m_V \to \infty$ limit, we can integrate out the vector mesons and get

$$H \to \frac{1}{24\pi^2}(R - L)^3 = S \implies B \to S , \tag{4.9}$$

as expected. In addition, it has been shown in [10] that $H$ and $S$ differ only by a total derivative term and therefore give rise to the same charge for any smooth configuration. Except for the definition of the current density, the physical difference between $S$ and $B$ comes when computing the baryon charge of singular configurations, such as the $N_f = 1$ baryons [10, 11]. In particular, for $N_f = 1$, the baryon current $B$ can be written as

$$B^{(N_f=1)} = -\frac{c_3 + c_4}{8\pi^2} d\omega d\eta' . \tag{4.10}$$

In section 3, VD led us to choose $c_3 = c_4 = 1$. With this choice, $B = 2H - S$ and at $N_f = 1$ it simplifies to

$$B^{(N_f=1)} = -\frac{1}{4\pi^2} d\omega d\eta' . \tag{4.11}$$

An example of a charged configuration is the $N_f = 1$ baryon introduced in [11]. This is a configuration constructed out of a finite $\eta' = \pi$ domain wall. On the boundary of the domain wall, $\eta'$ is not well defined and the chiral condensate is expected to vanish. $\eta'$ winds once around the ring that forms the boundary of the domain wall $\oint d\eta' = 2\pi$. Due to a conjectured emergent $U(1)_N$ CS theory on the domain wall, this system behaves as a boundary CS theory. The boundary is expected to carry edge modes, similar to the quantum Hall effect. The baryon charge of this configuration comes from the two orthogonal windings- the winding of $\eta'$ around the ring, and the winding of the CS field along the ring (the edge mode). (4.11) hints that the CS field on the DW is actually the $\omega$ vector meson. Indeed, configurations characterized by two orthogonal windings of the form

$$\oint \eta' = 2\pi\mathbb{Z} \; , \quad \oint \omega = 2\pi\mathbb{Z} \; , \tag{4.12}$$

have integer baryon charge under (4.11),[4]

$$-\frac{1}{4\pi^2} \int d\omega d\eta' \in \mathbb{Z} \; . \tag{4.13}$$

This quantization of charge fails to work for generic $c_{3,4}$ in (4.10). The charge is properly quantized for $c_3 + c_4 = 2$. In the next sections we will also show that the identification of $\omega$ as the $U(1)_N$ CS field on the $\eta' = \pi$ DW fixes $c_3 = 1$. The two demands together fix $c_3 = c_4 = 1$. Surprisingly, these are exactly the same parameters that give rise to VD.

# 5    Domain walls

In this section we will discuss QCD domain walls. Using the HLS Lagrangian with the hWZ action (2.12), we can study the DW theories when the vector mesons are included. Following [15], we will separate our discussion to $N_f = 1$ and $N_f \geq 2$. The different types of domain walls are described in the next sections.

## 5.1    $N_f = 1$ domain wall

In this section we will study domain wall configurations in $N_f = 1$ QCD with $\theta = \pi$ where we take the quark's mass $m$ to be real and positive. In the large $N$ small $m/\Lambda$ limits, we can write an effective Lagrangian for $\eta'$[21, 22]

$$\mathcal{L}_{\eta'} = \frac{F_\pi^2}{2}(\partial\eta')^2 + m\Lambda F_\pi^2 cos(\eta' + \theta) - \frac{1}{2}F_\pi^2 M_{\eta'}^2 \eta'^2 + \dots \; , \tag{5.1}$$

---

[4]In [10] it was assumed that the baryon current is $H$ which reduces at $N_f = 1$ to $H(N_f = 1) = -\frac{1}{8\pi^2}d\omega d\eta'$. This led to some confusion regarding the minimal charge and the validity of (4.12). Here we find that the correct current is actually $2H - S$, which solves the mentioned confusions.

with

$$F_\pi^2 \sim O(N) \ , \ \Lambda \sim O(1) \ , \ M_{\eta'}^2 \sim O(N^{-1}) \ . \tag{5.2}$$

For generic $\theta$ this potential has a unique global minimum, but for $\theta = \pi$ and $m > m_0 = \frac{M_{\eta'}^2}{\Lambda}$ there are two degenerate vacua related by time reversal symmetry $\eta' \to -\eta'$. The vacua are solutions to

$$m \sin(\eta') = m_0 \eta' \ . \tag{5.3}$$

For $m$ close to $m_0$ we can expand in small $\eta'$ and find the vacua

$$\eta_0'^2 \simeq 6(1 - m_0/m) \ . \tag{5.4}$$

On the other hand when $m_0 \ll m \ll \Lambda$, we can expand close to $\pm \pi$ and find

$$\eta_0' = \pm \pi (1 - m_0/m) + ... \ . \tag{5.5}$$

In the regime where (5.1) is valid, we see that as we increase $m$ (but keeping $m \ll \Lambda$), the vacua move from 0 towards $\pm \pi$. There are two topologically distinct trajectories that connect the two vacua. The first is the one that goes through $\eta' = 0$ and the second crosses the cusp at $\eta' = \pi$. It was argued in [15] that the tension coming from crossing the cusp is $T_{cusp} \sim N\Lambda^3$. Explicit classical computation shows that the tension of the domain wall that goes through $\eta' = 0$ is at most $\sim Nm^{\frac{1}{2}}\Lambda^{\frac{5}{2}}$ which is smaller than $T_{cusp}$ and therefore is energetically favorable. The theory on this domain wall can be extracted explicitly from (5.1) and is trivial (i.e. contains only the center of mass coordinate). Even though it is not the minimal tension configuration, the DW that goes through the cusp can still be considered as a metastable DW. The theory on it cannot be extracted from (5.1) due to the cusp. One of the things that can be done is to remove the cusp by integrating back in the fields that generate it. The cusp is a consequence of heavy fields that jump from one vacuum to the other. In the effective theory that contains both $\eta'$ and the heavy fields, the cusp doesn't appear in the Lagrangian and the emergent theory on the $\eta' = \pi$ DW can be read off directly from this effective theory. According to a conjecture made in [10], these fields are the vector mesons. We will elaborate more about this conjecture and the motivation behind it in section 6. For now, we will simply assume that this conjecture is correct. In that case, the theory on the DW can be read off from the effective theory of $\xi_{R,L}$ and $V$. In this formalism, the relevant domain wall configuration is of the form

$$\eta' = \eta'(z) \ , \ \lim_{z \to -\infty} \eta'(z) = 0 \ , \ \lim_{z \to +\infty} \eta'(z) = 2\pi \ . \tag{5.6}$$

Plugging it into the $N_f = 1$ version of (2.11), we get on the domain wall (after throwing a full derivative term)

$$\mathcal{L}_{DW} = \frac{Nc_3}{4\pi} \omega d\omega \ . \tag{5.7}$$

in addition, there is a mass term for $\omega$ coming from (2.9). We see that by choosing $c_3 = 1$, the theory on this domain wall is $U(1)_N$ Chern-Simons-Higgs theory. Surprisingly,

$c_3 = 1$ is the value predicted by VD. The emergence of an $U(1)_N$ CS theory on the $\eta' = \pi$ DW is also needed for the correct construction of the $N_f = 1$ baryon [11] as explained in section 4.

This is valid in the $m \ll \Lambda$ limit. We can also study the opposite limit $\Lambda \ll m$ which is approximately pure YM at $\theta = \pi$. The domain wall in this case is [20] $U(1)_N \simeq SU(N)_{-1}$ pure CS theory. An interesting question is how the two limits are connected. In [15] it was suggested that the theory on the DW can be described for every value of $m$ as $U(1)_N$ CS theory coupled to one fundamental scalar. When the mass of the scalar is very large and positive, we can integrate it out and the theory is simply $U(1)_N$ as in pure YM. When the mass of the scalar is very large and negative we are in the Higgs phase which is gapped at low energies. The authors of [15] identified this Higgs phase as the theory on the trivial DW that goes through $\eta' = 0$. We suggest to slightly modify their proposal and identify the Higgs phase as the theory on the DW that goes through the cusp (5.7). This modification means that the pure YM DW is continuously connected to the metastable DW at small mass which implies a first order phase transition between the two domain walls. If this is correct, than there should be a critical mass $m_c \sim \Lambda$ in which the two tensions are equal. Except for the appearance of a $U(1)_N$ CSH theory (5.7) on the DW when crossing the cusp, the existence of a phase transition can be motivated as follows. First, we see that the tension difference between the two domain walls decreases as the mass increases which makes the scenario of phase transition plausible (see figure 1). When $\eta'_0 \to \pm\pi$, the tension of the DW that goes through the cusp is not expected to depend on $m$, unlike the one that goes through $\eta' = 0$. Therefore, when the mass is very large, the tension of the cusped DW is expected to remain finite, while the tension of the $\eta' = 0$ DW is expected to diverge. This makes it very plausible that a phase transition between the two DWs indeed happens. Another argument comes from the relation between shifts of $\theta$ to shifts of $\eta'$ and the cusp at $\theta = \pi$ to the cusp at $\eta' = \pi$. The pure YM DW exists on the cusp at $\theta = \pi$. We can also think of it as an interface interpolating from $\theta = \pi - \epsilon$ on one side to $\theta = \pi + \epsilon$ on the other side. In the $\epsilon \to 0$ limit, this interface is exactly the $\theta = \pi$ DW. Similarly, the two vacua of the $\eta'$ potential asymptote to $\eta' = \pm\pi$ as the mass is taken to be large. Even though we know what happens only in the two limits $m \ll \Lambda$, and $m \gg \Lambda$, the most natural picture is that the pure YM DW is connected continuously to the DW that goes through the cusp. The theory on this DW can be described as a $U(1)_N$ CS theory that flows smoothly from the topological phase at $m \to \infty$ to the Higgs phase at $m \to 0$. This proposal has several interesting consequences. According to this proposal, the mass of the $\omega$ vector meson on the $\theta = \pi$ DW goes to zero as one increases the mass of the quark. Moreover, when $m \to \infty$, $\omega$ should become massless on the DW and transform continuously into some kind of a glueball. As explained in section 1, the fact that there exists a continuous deformation of $\omega$ which is massless and should have a description in terms of gluons motivates the idea of Seiberg-like duality between vector mesons and gluons also above the IR.

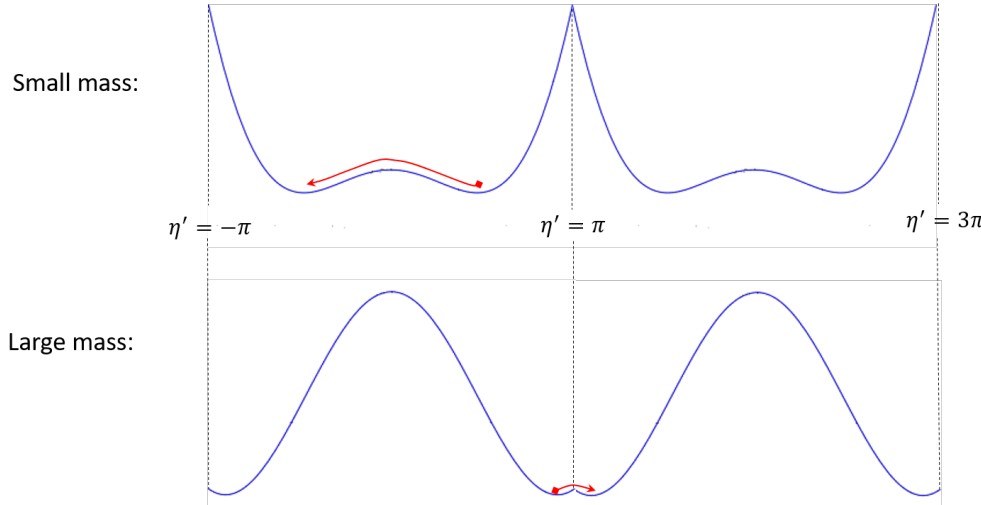

Figure 1: The $\eta'$ potential at $\theta = \pi$ for small and large values of the masses. At small mass, it is energetically favorable to avoid the cusp. As the mass increases, the tension difference between the two domain walls decreases and we conjecture that above some critical mass $m > m_c \sim \Lambda$ it is favorable to cross the cusp. Therefore the cusped DW is connected continuously to the YM DW.

## 5.2 The pionic $N_f \geq 2$ domain wall

In this section we will study the $N_f \geq 2$ pionic domain wall. The theory we study is $\theta = \pi$ QCD with $N_f \geq 2$ and equal small mass for all the quarks $0 < m \ll \Lambda$. For simplicity, we will also ignore $\eta'$ as it is not going to play any role in this domain wall. As was shown in [15], this theory has two degenerate vacua related by time reversal given by

$$U_1 = 1 \ , \ U_2 = e^{-2\pi i/N_f} \ . \tag{5.8}$$

We can impose the boundary conditions

$$\lim_{z \to -\infty} U = 1 \ , \ \lim_{z \to \infty} U = e^{-2\pi i/N_f} \ . \tag{5.9}$$

The minimal tension configuration satisfying the boundary conditions is of the form

$$U_{DW} = \begin{pmatrix} e^{i\alpha} \\ & e^{i(1-N_f)\alpha} \end{pmatrix} \ , \ \lim_{z \to -\infty} \alpha(z) = 0 \ , \ \lim_{z \to +\infty} \alpha(z) = -\frac{2\pi}{N_f} \ , \tag{5.10}$$

where the first entry in $U_{DW}$ represents an $(N_f - 1) \times (N_f - 1)$ block. This configuration breaks $SU(N_f)_V \to S[U(N_f - 1) \times U(1)]$. Therefore, the theory on the domain wall contains a $\mathbb{CP}^{N_f-1} = \frac{SU(N_f)}{S[U(N_f-1) \times U(1)]}$ sigma model. In addition, the sigma model inherits a level $N$ WZ term from the 4d WZ term. The authors of [15] identified this theory with the low energy limit of $U(1)_N$ CS theory coupled to $N_f$ fundamental scalars in the Higgs phase. The sigma model is a result of the same symmetry breaking

pattern $SU(N_f) \to S[U(N_f - 1) \times U(1)]$, and the WZ term comes from the CS term when integrating out the vector fields. There is a conjectured dual description to this theory in terms of $SU(N)_{N_f/2-1}$ CS coupled to $N_f$ fundamental fermions [24]. These theories

$$U(1)_N + N_f \ \phi \simeq \ SU(N)_{N_f/2-1} + N_f \ \psi \tag{5.11}$$

in the large mass limit (positive for scalars, negative for the fermions in this convention) become

$$U(1)_N \simeq SU(N)_{-1} \tag{5.12}$$

which is the $\theta = \pi$ domain wall theory in pure YM [20]. Therefore, they proposed that (5.11) describes the theory on the domain wall for every value of the quarks mass. Using the HLS Lagrangian together with the hWZ action, we can find the DW theory in a regime where the vector fields are treated as dynamical. As we will see, this results in the $\mathbb{CP}^{N_f-1}$ sigma model coupled to $U(N_f - 1)$ massive vector fields. The emergent $U(N_f - 1)$ CS term on the DW contributes to the WZ term of the sigma model, in contradiction to the proposal of [15]. Their conjecture was based on the assumption that the $\mathbb{CP}^{N_f-1}$ at small mass is connected continuously to the pure YM DW at $m \to \infty$, and therefore the DW theory should be described by a 3d DW theory with the two phases on the edges of its parameter space. As in 5.1, we will argue that there is a first order phase transition between topologically distinct DWs, and that the YM DW is not connected continuously to the $\mathbb{CP}^{N_f-1}$ DW but to some metastable DW at small mass. We will start from analyzing the kinetic terms of the HLS Lagrangian

$$\mathcal{L}_2 = \frac{F_\pi^2}{4} tr(\partial U \partial U^\dagger) - \frac{a F_\pi^2}{4} tr(\partial \xi_R \xi_R^\dagger + \partial \xi_L \xi_L^\dagger - 2iV)^2 \ . \tag{5.13}$$

A general expansion around (5.10) that doesn't change the boundary conditions can be written as

$$U = g U_{DW} g^\dagger \ , \ g \in SU(N_f) \ . \tag{5.14}$$

At low energies we can expand

$$g = 1 + i \begin{pmatrix} 0 & \sigma \\ \sigma^\dagger & 0 \end{pmatrix} - \frac{1}{2} \begin{pmatrix} \sigma \sigma^\dagger & 0 \\ 0 & \sigma^\dagger \sigma \end{pmatrix} + O(\sigma^3) \ , \tag{5.15}$$

where $\sigma(x^\mu \neq z)$ is an $N_f - 1$ dimensional vector parametrizing the coordinates on the $\mathbb{CP}^{N_f-1}$ manifold around (5.10). The first kinetic term on the DW becomes

$$\frac{F_\pi^2}{4} \int dz tr(\partial U \partial U^\dagger) = 2F_\pi^2 \int dz sin^2(N_f \alpha/2) \partial \sigma^\dagger \partial \sigma + O(\sigma^4) \ . \tag{5.16}$$

To study the second kinetic term, we first need to find a good expansion for $\xi_{R,L}$ and $V$. Demanding that $U = \xi_L^\dagger \xi_R$ we can write in general

$$\xi_R = h \xi_{DW} g^\dagger \ , \ \xi_L = h \xi_{DW}^\dagger g^\dagger \ , \ \xi_{DW} = U_{DW}^{\frac{1}{2}} \ . \tag{5.17}$$

This implies

$$\partial_\mu\xi_R\xi_R^\dagger + \partial_\mu\xi_L\xi_L^\dagger = h[\xi_{DW}\partial g^\dagger g\xi_{DW}^\dagger + \xi_{DW}^\dagger\partial g^\dagger g\xi_{DW} - 2\partial h^\dagger h]h^\dagger$$

$$= h\left[-2i\cos(N_f\alpha/2)\begin{pmatrix} 0 & \partial\sigma \\ \partial\sigma^\dagger & 0 \end{pmatrix} - 2\partial h^\dagger h\right]h^\dagger + O(\sigma^2)\ . \tag{5.18}$$

A convenient choice for $h$ is the gauge in which the linear term $O(\sigma)$ in (5.18) vanishes. This is achieved by choosing

$$h = 1 + i\cos(N_f\alpha/2)\begin{pmatrix} 0 & \sigma \\ \sigma^\dagger & 0 \end{pmatrix} - \frac{1}{2}\cos^2(N_f\alpha/2)\begin{pmatrix} \sigma\sigma^\dagger & \\ & \sigma^\dagger\sigma \end{pmatrix} + ...$$

$$\Rightarrow \ \partial h^\dagger h = -i\cos(N_f\alpha/2)\begin{pmatrix} 0 & \partial\sigma \\ \partial\sigma^\dagger & 0 \end{pmatrix} + \frac{1}{2}\cos^2(N_f\alpha/2)\begin{pmatrix} \partial\sigma\sigma^\dagger - \sigma\partial\sigma^\dagger & \\ & \partial\sigma^\dagger\sigma - \sigma^\dagger\partial\sigma \end{pmatrix}\ . \tag{5.19}$$

With this choice, we get

$$h^\dagger(\partial_\mu\xi_R\xi_R^\dagger + \partial_\mu\xi_L\xi_L^\dagger)h = \sin^2(N_f\alpha/2)\begin{pmatrix} \partial\sigma\sigma^\dagger - \sigma\partial\sigma^\dagger & \\ & \partial\sigma^\dagger\sigma - \sigma^\dagger\partial\sigma \end{pmatrix} + O(\sigma^3)\ . \tag{5.20}$$

In addition, we take the ansatz $V = f(z)\begin{pmatrix} v_a & v_b \\ v_b^\dagger & v_c \end{pmatrix}$, where $\lim_{z\to\pm\infty} f(z) = 0$ and $v_{a,b,c}$ are independent of $z$. $f(z)$ can in principle be found by solving some differential equations. However, the exact form of $f(z)$ will not be important for us since after integrating over $z$, the effect of changing $f(z)$ will just result in a different numerical factor that can be swallowed into the 3d gauge coupling on the DW. Therefore, we allow ourselves to take for simplicity $f(z) = \sin^2(N_f\alpha/2)$ which gives the nicest form for the Lagrangian even though it is not correct. With this ansatz we get

$$tr(\partial\xi_R\xi_R^\dagger + \partial\xi_L\xi_L^\dagger - 2iV)^2 = \sin^4(N_f\alpha/2)tr\begin{pmatrix} \partial\sigma\sigma^\dagger - \sigma\partial\sigma^\dagger - 2iv_a & -2iv_b \\ -2iv_b^\dagger & \partial\sigma^\dagger\sigma - \sigma^\dagger\partial\sigma - 2iv_c \end{pmatrix}^2$$

$$= -8\sin^4(N_f\alpha/2)v_b^\dagger v_b + \sin^4(N_f\alpha/2)[(\partial\sigma\sigma^\dagger - \sigma\partial\sigma^\dagger - 2iv_a)^2 + (\partial\sigma^\dagger\sigma - \sigma^\dagger\partial\sigma - 2iv_c)^2]\ . \tag{5.21}$$

We see that $v_b$ and $tr(V) = tr(v_a) + v_c$ decouple so we can simply ignore them and set them to zero. A simpler ansatz can then be used $V = \sin^2(N_f\alpha/2)\begin{pmatrix} v & \\ & -tr(v) \end{pmatrix}$ with $v \in u(N_f - 1)$. The entire kinetic term (5.13) on the DW is

$$2F_\pi^2\int dz\left[\sin^2(N_f\alpha/2)\partial\sigma^\dagger\partial\sigma + \frac{a}{2}\sin^4(N_f\alpha/2)[tr(v^2) + tr(v)^2 + itr((v + tr(v))(\partial\sigma\sigma^\dagger - \sigma\partial\sigma^\dagger))]\right] + ...\ . \tag{5.22}$$

The ... contain higher order terms. The novel part is the interactions between $\sigma$ and the $U(N_f - 1)$ vector fields so we included them even though they are of order $O(\sigma^4)$ which we threw.

Now we are ready to deal with the WZ term (2.1)+(2.11). The details of the computation appear in appendix A and result in the emergent action on the DW,

$$\frac{9N(c_1 - c_2) + 3Nc_3 - 8N}{32\pi}\sigma^\dagger d\sigma d\sigma^\dagger d\sigma + \frac{3iN[2(1 - 2N_f)(c_1 - c_2) - c_3]}{64\pi N_f}d\sigma^\dagger d\sigma tr(v)$$
$$- \frac{3iN[2(1 + N_f)(c_1 - c_2) + (N_f - 1)c_3]}{64\pi N_f}d\sigma^\dagger v d\sigma - \frac{3Nc_3}{64\pi N_f}[tr(vdv) + (1 - N_f)tr(v)tr(dv)] \ .$$
$$(5.23)$$

For the specific choice of $c_1 - c_2 = c_3 = 1$ we get

$$\frac{N}{8\pi}\sigma^\dagger d\sigma d\sigma^\dagger d\sigma + \frac{3iN(1 - 4N_f)}{64\pi N_f}d\sigma^\dagger d\sigma tr(v) - \frac{3iN(1 + 3N_f)}{64\pi N_f}d\sigma^\dagger v d\sigma$$
$$- \frac{3N}{64\pi N_f}[tr(vdv) + (1 - N_f)tr(v)tr(dv)] + O(\sigma^6) \ .$$
$$(5.24)$$

One can check that if we integrate out the vector fields by plugging $v = -id\sigma\sigma^\dagger + ...$ we get the required level N WZ term,

$$-\frac{N}{4\pi}\sigma^\dagger d\sigma d\sigma^\dagger d\sigma + ... \ . \tag{5.25}$$

We can ask about the interpretation of this result. The full DW theory, whatever it is, should be consistent with the above 3d Lagrangian. This excludes for example the conjecture of [15] because it doesn't contain any $U(N_f - 1)$ vector fields. According to their conjecture, the level N WZ term on the DW comes from integrating out a Higgsed $U(1)_N$ vector field, which is not what we get here. A natural guess can be $U(N_f - 1)$ gauge theory coupled to $N_f$ fundamental scalars. Assuming that in the deep Higgs phase the vacuum falls into the maximally flavor-color locking phase [25], the low energy theory has an expansion as $\mathbb{CP}^{N_f - 1}$ sigma model interacting with massive $U(N_f - 1)$ vector fields. However there are several problems with this identification. The first is that the separation between $tr(v)$ and $v$ in our expansion is different than what expected from the above proposal. This problem is already at the level of the kinetic term and doesn't depend on the details of the topological term. The second problem is that it looks like the induced CS term has a fractional level and that except for the CS term, the other terms don't seem to come from any reasonable (and not irrelevant) term in the uv. In any case, there is no good reason why a description in terms of some 3d uv complete theory should even exist. The 4d parent theory by itself is not described as a uv complete theory but as a non-linear sigma model. As will be explained in more detail in the next section, the fractional level is not really a problem since the entire theory is gauge invariant.

## 5.3   $N_f \geq 2$ $\eta'$ domain wall

In this section we will study the domain wall configuration in $N_f \geq 2$ QCD that goes only through the $\eta'$. At small quarks' masses the minimal tension DW is the pionic

DW studied in the previous section. However, the $\eta'$ DW can still be studied at least as a metastable DW. We will impose the same boundary conditions as in the previous section (5.9) and consider the configuration

$$U = e^{\frac{i}{N_f}\eta'(z)} \ , \ \lim_{z\to-\infty} \eta'(z) = 0 \ , \ \lim_{z\to+\infty} \eta'(z) = 2\pi \ . \tag{5.26}$$

As in 5.1, we will remove the $\eta' = \pi$ cusp by integrating in the vector mesons. The hWZ action generates on the DW the following term,

$$\begin{aligned}
\mathcal{L}_{DW,\eta'} &= \frac{N}{24\pi N_f}[R_D^3 + L_D^3 + 3iF(R_D + L_D)] \\
&= \frac{N}{4\pi N_f}\left[VdV - \frac{2i}{3}V^3\right] + \frac{N}{24\pi N_f}[R^3 + L^3 + 3id(V(R+L))] \ .
\end{aligned} \tag{5.27}$$

In addition, there is a mass term for the vector mesons coming from the HLS Lagrangian as usual. Notice that the second term in the last line contains a full derivative $d(V(R+L))$ and the term $R^3+L^3$ which vanishes in the unitary gauge. Ignoring these terms, the theory on the DW is a $U(N_f)_{N/N_f}$ CSH theory. This theory is of course not properly quantized and inconsistent by itself for generic $N, N_f$. However, the full theory is gauge invariant as can be seen explicitly from the first line of (5.27).

As in 5.1, we can ask here which one of the domain walls is connected continuously to the pure YM DW. The same arguments of 5.1 hold also here and we therefore conjecture that it is (5.26) that is connected continuously to the YM DW. This DW is metastable when the mass of the quarks is small and there should be a phase transition between the two domain walls at some critical mass $m_c \sim \Lambda$. We can start from small $m$ and continuously take it to infinity. Assuming our conjecture is correct, we expect the following flow

$$"U(N_f)_{N/N_f}" \to_{m\to\infty} U(1)_N \ , \tag{5.28}$$

where $"U(N_f)_{N/N_f}"$ should be understood as (5.27). This can happen in the following way. When we take the mass of the quarks to infinity, generically all the mesons will decouple and disappear from the spectrum. In section 5.1, we conjectured that the mass of the $\omega$ meson on the DW actually goes to zero as the quark's mass goes to infinity at $\theta = \pi$. If this is true, the same thing happens here. We will assume that only the trace of $V$ remains dynamical as the mass increases, such that $V \to w\mathbb{1}$ where $w$ becomes in this limit a properly normalized $U(1)$ gauge field. Then, (5.27) flows to

$$\frac{N}{4\pi N_f}tr\left(VdV - \frac{2i}{3}V^3\right) + ... \to \frac{N}{4\pi N_f}tr\,(wdw) = \frac{N}{4\pi}wdw \ , \tag{5.29}$$

which is the desired $U(1)_N$ CS theory. An important check of our proposal is to verify that the DW theory reproduces the correct anomaly associated with $U(N_f)/\mathbb{Z}_N$ when $gcd(N, N_f) \neq 1$[26, 27]. This analysis will be done elsewhere [23]. Another interesting question is whether $"U(N_f)_{N/N_f}"$ has some kind of dual description in the spirit of $SU(N)$ CS theory coupled to fermions. We leave this to future work.

# 6 The cusp potential and a gluons-mesons duality

In this section we will also take the large $N$ limit in which the matrices $U, \xi_{R,L}$ and the vector fields $V$ are $U(N_f)$ valued. When $N \to \infty$, $U(1)_A$ becomes a good symmetry of the theory. As a result, $\eta'$, which we define as $e^{i\eta'} \equiv det\ U$, becomes a massless NG boson associated with the breaking of $U(1)_A$. The potential for the $\eta'$ is suppressed in the large $N$ limit. The leading part in its potential takes the form, [21, 22]

$$V_{\eta'} = \frac{1}{2}F_\pi^2 M_{\eta'}^2 \min_{k \in \mathbb{Z}}(\eta' - 2\pi k)^2 + ... \ , \tag{6.1}$$

where $F_\pi^2 \sim N$, $M_{\eta'}^2 \sim \frac{1}{N}$.[5] The potential is locally quadratic but has a cusp whenever $\eta' = \pi \mod 2\pi$. The cusp represents a phase transition between two branches. As one crosses the cusp, some heavy fields jump from one vacuum to another. This behavior has several important consequences:

1. It is possible to eliminate the cusp by "integrating in" the heavy fields that generated it. In other words, in the effective theory that contains both $\eta'$ and those heavy fields, the potential for $\eta'$ doesn't appear explicitly in the Lagrangian.

2. Any emergent CS theory that might appear on $\eta' = \pi$ wall (see section 5) or disc (see section 4) comes from those heavy fields.

3. Combining the two, we can conclude that it should be possible to extract the CS theory directly from the effective theory of both $\eta'$ and the heavy fields.

The conventional picture for the formation of the cusp potential was described in [22] (see also [28]). We will start by reviewing it and show that it indeed satisfies the above properties. Consider the gluonic topological density

$$q = \frac{1}{8\pi^2}tr\ G \wedge G \ , \tag{6.2}$$

where $G$ is the field strength for the gluons. We can write an effective action for $q$ and $\eta'$. The coupling between $\eta'$ and $q$ is fixed from the $U(1)_A$ axial anomaly, which implies[6]

$$\eta' \to \eta' + \alpha \ \Rightarrow \ \mathcal{L} \to \mathcal{L} + \alpha q \ . \tag{6.3}$$

This is satisfied by writing

$$\mathcal{L}_{q\eta'} = -\frac{i}{2}qtr(log(U) - log(U^\dagger)) = \eta'q \ . \tag{6.4}$$

---

[5]This is $\frac{1}{N}$ suppressed with respect to the kinetic term $\frac{1}{2}F_\pi^2(\partial\eta')^2$.

[6]Notice a sign difference from [22]. There are several conventions for how $U$ is defined in terms of its transformation law under the global symmetries. (6.3) is consistent with the convention in which the WZ term is defined with a minus sign as in (2.1).

We can also add some general function of $q$, but in the large $N$ limit, higher order terms are suppressed and only the quadratic term $q^2$ survives. The two terms together give the effective theory for $q$,

$$\mathcal{L}_q = \frac{1}{2F_\pi^2 M_{\eta'}^2} q^2 + \eta' q . \qquad (6.5)$$

At this point we can integrate $q$ out. Locally, we can use the equation of motion, $q = F_\pi^2 M_{\eta'}^2 \eta'$ and get

$$\mathcal{L}_q \rightarrow -\frac{F_\pi^2 M_{\eta'}^2}{2} \eta'^2 . \qquad (6.6)$$

Globally, we must also impose the $2\pi$ periodicity of $\eta'$ by taking into account the quantization of the instanton number $\int q \in \mathbb{Z}$. Demanding this we get (6.1).

Except for generating the cusp, (6.4) is also responsible for CS terms on $\eta' = \pi$ walls. This can be seen by writing it as

$$\frac{1}{8\pi^2} \eta' G \wedge G = -\frac{1}{2\pi} d\eta' \wedge \frac{1}{4\pi} \left( AdA - \frac{2i}{3} A^3 \right) + d(...) , \qquad (6.7)$$

where we see that $\frac{1}{2\pi} d\eta'$ is coupled to an $SU(N)_{-1}$ CS term with gluons as the vector fields.

It is interesting to compare the coupling (6.7) to the way $\eta'$ couples to the vector mesons via the hWZ action. While the $\eta'$ doesn't enter into the regular WZ term (2.1), it does appear explicitly in the hWZ term (2.12). Lets start from the case of $N_f = 1$. In that case, only $\mathcal{L}_3$ in (2.11) survives and we get (ignoring a total derivative term)

$$\mathcal{L}_{hWZ} = -\frac{Nc_3}{8\pi^2} \eta' d\omega d\omega = \frac{1}{2\pi} d\eta' \wedge \frac{c_3 N}{4\pi} \omega d\omega . \qquad (6.8)$$

Here, $\frac{1}{2\pi} d\eta'$ is coupled to a $U(1)_{c_3 N}$ CS term with $\omega$ as the vector field. For $c_3 = 1$, the two CS theories are dual to each other,[7]

$$SU(N)_{-1} \simeq U(1)_N . \qquad (6.9)$$

As we already mentioned, $c_3 = 1$ is exactly the choice implied by VD and the required value for the consistent construction of $N_f = 1$ baryons.

What about $N_f \geq 2$? For general $\{c_i\}$, $\eta'$ interacts with vector mesons and pions in quite a complicated way,

$$\mathcal{L}_{\eta'} = \frac{Nc_1}{32\pi^2 N_f} d\eta' [LR^2 + RL^2 - 2iV(RL + LR) - 4iV(R^2 + L^2) - 6V^2(R + L) + 4iV^3]$$

$$+ \frac{Nc_2}{16\pi^2 N_f} d\eta' [L^2 R + R^2 L - 3V^2(R + L) - iV(R^2 + L^2) - 2iV(RL + LR) + 2iV^3]$$

$$+ \frac{iNc_3}{16\pi^2 N_f} d\eta' [V(R^2 + L^2) - 2iVdV - iV^2(R + L) - 2V^3] . \qquad (6.10)$$

_______________________________

[7]To be more precise, the theory on the mesonic side is $U(1)_N$ CSH theory. Similarly, the theory on the gluonic side is $SU(N)_{-1}$ strongly interacting with fermions in some way. So the duality is really some version of $SU(N)_{-1} + fermions \simeq U(1)_N + scalars$. The same story holds also for (6.12).

However, given $c_3 = 1$, there is a unique choice $c_2 = -\frac{c_3}{3}$ , $c_1 = \frac{2c_3}{3}$ that results in the nice coupling,[8]

$$\mathcal{L}_{\eta'} = -\frac{N}{8\pi^2 N_f}\eta' F^2 = \frac{1}{2\pi N_f}d\eta' \wedge \frac{N}{4\pi}\left(V dV - \frac{2i}{3}V^3\right) \ . \tag{6.11}$$

Again, this choice of parameters is consistent with VD. Generalizing (6.9), we see that $\frac{1}{2\pi N_f}d\eta'$ is coupled to an $SU(N)_{-N_f}$ CS term in (6.7), and to $U(N_f)_N$ CS term in (6.11) which are again dual to each other

$$SU(N)_{-N_f} \simeq U(N_f)_N \ . \tag{6.12}$$

One might be bothered about the factor of $N_f$ in the denominator of (6.11). The motivation behind looking at $\frac{d\eta'}{2\pi N_f}$ is that $U$ is invariant under $\eta' \to \eta' + 2\pi N_f$. However, it looks like on $\eta' = \pi$ walls, the emergent theory will have a fractional level $U(N_f)_{N/N_f}$. This issue was already explained in section 5.3 when studying the related DW in more detail.

The resemblance between (6.7) and (6.11) and the CS duality motivates the idea that (6.7) and (6.11) are actually two equivalent descriptions related by duality. This duality can be interpreted as a consequence of a conjectured Seiberg-like duality between gluons and vector mesons,[9, 10, 17–19] as explained in the introduction. Assuming that (6.11) is indeed dual to (6.7), it means that instead of using the mechanism of [22], we can repeat the entire story using vector mesons. In particular, it means that the cusp potential is generated by integrating out the vector mesons. The details of how it happens will be studied elsewhere [23]. In this paper, we simply assume that this is correct and that when including the vector mesons as dynamical fields, the $\eta'$ potential is absent. This assumption is crucial in the identification of the vector mesons as the CS fields used throughout the paper. The reason is that if the assumption is not correct, the hWZ action and (6.7), (or equivalently, the hWZ action and the cusp) exist together side by side in the same description. Then, the theory on the $\eta' = \pi$ DW will get two independent contributions, one coming from the cusp and the other from the hWZ action. The theory on the DW will then be very roughly $SU(N)_{-1} + U(1)_N$ (in $N_f = 1$ for simplicity), and using the $SU(N)_{-1} \simeq U(1)_N$ duality, we will have two different $U(1)_N$ CS fields, one which is the $\omega$ meson, and the second which is some emergent field on the DW. While we cannot exclude this possibility, it looks too much of a coincidence. The most plausible scenario to our opinion is that the duality between vector mesons and gluons indeed holds. Then, the formation of the cusp and the DW theory have two dual descriptions as explained above.

# 7    Summary of $c_i$ conditions and finite $N$

In this work, we showed how the three concepts- vector dominance, baryon symmetry, Chern-Simons theory- are related and originate from the same hidden Wess-Zumino

---

[8]This is after throwing away some total derivative terms. See (5.27) for the full expression.

action. The relation holds in the large $N$ limit where the action can be written as single trace in flavor space over $U(N_f)$ valued matrices. As was shown, the most general hWZ action coupled to external $U(1)$ vector-like gauge field contains 4 free real parameters, $c_i$ , $i = 1, ..., 4$. The first theoretical constraints are based on $N_f = 1$ physics. By requiring that the $N_f = 1$ baryon will have charge 1 under the baryon charge, we found $c_3 + c_4 = 2$. By requiring that the theory on the $\eta' = \pi$ DW at $N_f = 1$ will contain a $U(1)_N$ CS term (which is also a crucial ingredient in the construction of the $N_f = 1$ baryon), we found $c_3 = 1$. These two demands are equivalent to the vanishing of the two vertices $AA\Pi$ and $AV\Pi$. Finally, by requiring that when adding more flavors, the coupling of $d\eta'$ to the vector mesons will still have the structure of a CS term (or equivalently, the coupling of $\eta'$ to the vector mesons will have the form of an axion coupling), we found $c_1 = \frac{2c_3}{3}$ , $c_2 = -\frac{c_3}{3}$, which is consistent with the vanishing of the vertex $V\Pi\Pi\Pi\Pi$. These theoretical demands fix completely the hWZ action and imply VD. This relation gives a theoretical explanation of the phenomenological principle of VD. The conditions on $c_{1,2}$ may seem less motivated than the ones on $c_{3,4}$. In any case, even if one throws them away, most of the results of this paper are not affected.

At finite $N$, one can add multi-trace operators that distinguish between the $SU(N_f)$ and the $U(1)$ parts and break the triple relation. This type of terms can be used to explain possible violations of VD in the real world. The term that should remain as it is also at finite $N$, is

$$-\frac{N}{8\pi^2}d\eta'\omega d\omega \ . \tag{7.1}$$

This can provide a non-trivial check of our conjecture by measuring the coefficient of this term in real world QCD.[9] Another type of corrections comes when interpreting the "h" in hWZ as standing for "homogeneous". This means that we assumed that the coefficients $c_i$ don't depend on the so-called dilaton field, which is roughly speaking the radius of the target space [29–31]. Corrections to the hWZ action that depend on the dilaton may serve as another source for discrepancies between the measured hWZ action at low energies to the predicted hWZ action at higher energies [14]. In particular for the $N_f = 1$ baryons close to the singular ring, these type of corrections might be quite important. We ignore these corrections completely in this work and leave it as an open problem.

# Acknowledgments

We would like to thank Pietro Benetti Genolini, Philip Boyle Smith, Joe Davighi, Ryuichiro Kitano, Nakarin Lohitsiri, Ryutaro Matsudo, Carl Turner, Shimon Yankielowicz for many useful discussions. We would also like to especially thank Zohar Komargodski and David Tong for many discussions, insights and going over the draft, and

---

[9]Notice that our definitions for $\eta'$ and $\omega$ as the $U(1) \in U(N_f)$ fields differ from their definitions in real world QCD, where $\eta'$ is the $U(N_f = 3)$ singlet and $\omega$ is the $U(N_f = 2)$ singlet.

the Blavatnik family foundation for the generous support. A.K is supported by the Blavatnik postdoctoral fellowship and partially by David Tong's Simons Investigator Award. This work has been also partially supported by STFC consolidated grant ST/P000681/1.

# A    Some computations

In this appendix, we will compute the emergent action on the $N_f \geq 2$ pionic DW described in section 5.2 coming from the hWZ action. We will write explicitly the results up to order $O(\sigma^4)$ where $V \sim O(\sigma^2)$. We will start from the regular WZ term (2.1). This computation also appeared in [32]. We consider the DW configuration

$$U = g U_{DW} g^\dagger \,, \tag{A.1}$$

as in (5.14) and expand

$$g = 1 + i \begin{pmatrix} 0 & \sigma \\ \sigma^\dagger & 0 \end{pmatrix} - \frac{1}{2} \begin{pmatrix} \sigma\sigma^\dagger & 0 \\ 0 & \sigma^\dagger\sigma \end{pmatrix} + \dots \,. \tag{A.2}$$

The action becomes

$$\frac{iN}{48\pi^2} \int_{\mathcal{B}_5} \partial_z UU^\dagger (dUU^\dagger)^4 = \frac{N_f N}{3\pi^2} \int_{\mathcal{B}_5} \partial_z\alpha \sin^4(N_f\alpha/2) d\sigma^\dagger d\sigma d\sigma^\dagger d\sigma + O(\sigma^6) \,, \tag{A.3}$$

where $d$ now is the derivative in the directions transverse to $z$, and we used

$$\partial_z UU^\dagger = i\partial_z\alpha gTg^\dagger \,, \quad T = \begin{pmatrix} 1 & \\ & 1 - N_f \end{pmatrix} \,, \quad dUU^\dagger = i \begin{pmatrix} & (1 - e^{iN_f\alpha})d\sigma \\ (1 - e^{-iN_f\alpha})d\sigma^\dagger & \end{pmatrix} + \dots. \tag{A.4}$$

The $z$ integration can be evaluated using

$$\int_{-\infty}^{\infty} dz\partial_z\alpha \sin^4(N_f\alpha/2) = \int_0^{-2\pi/N_f} d\alpha \sin^4(N_f\alpha/2) = -\frac{3\pi}{4N_f} \,. \tag{A.5}$$

The emergent 3d term is then

$$-\frac{N}{4\pi}\sigma^\dagger d\sigma d\sigma^\dagger d\sigma + O(\sigma^6) \,, \tag{A.6}$$

which is indeed a level $N$ WZ term. Now we will perform the computation for the hWZ action (2.11). $V_z$ appears explicitly in (2.11) and we need find its value on the DW. The bulk kinetic term in the $z$ direction is proportional to (at leading order)

$$tr(R_z + L_z - 2iV_z)^2 = tr\left( -iN_f\partial_z\alpha \sin(N_f\alpha/2) \begin{pmatrix} & \sigma \\ \sigma^\dagger & \end{pmatrix} - 2iV_z \right)^2 \,. \tag{A.7}$$

$V_z$ is a scalar on the DW and for simplicity we can plug in

$$V_z = -\frac{N_f}{2}\partial_z\alpha sin(N_f\alpha/2)\begin{pmatrix} & \sigma \\ \sigma^\dagger & \end{pmatrix} , \tag{A.8}$$

which is correct at low energies. Using the expansions of $R, L, V$ on the DW as written in section 5.2, we can write at leading order

$$R_D = I_0 + I_1 , \quad L_D = -I_0 + I_1 , \tag{A.9}$$

with

$$I_{0,z} = \frac{i}{2}\partial_z\alpha T , \quad I_{1,z} = 0 , \quad I_{0,\perp} = sin(N_f\alpha/2)\begin{pmatrix} & d\sigma \\ -d\sigma^\dagger & \end{pmatrix} ,$$

$$I_{1,\perp} = sin^2(N_f\alpha/2)\begin{pmatrix} d\sigma\sigma^\dagger - iv & \\ & -\sigma^\dagger d\sigma + itr(v) \end{pmatrix} . \tag{A.10}$$

It is straight forward to check that $I_0^4 = 0$ , $I_1 I_0^3 \sim (\sigma^4)$ and all the other combinations are of higher order. Therefore, it is enough to keep the $I_1 I_0^3$ terms from $\mathcal{L}_{1,2}$, which are

$$\frac{Nc_1}{16\pi^2}\mathcal{L}_1 = \frac{iNc_1}{4\pi^2}I_1 I_0^3 + O(\sigma^6) , \quad \frac{Nc_2}{16\pi^2}\mathcal{L}_2 = -\frac{iNc_2}{4\pi^2}I_1 I_0^3 + O(\sigma^6) . \tag{A.11}$$

After performing the integration over $z$ as in (A.5), we get on the DW,

$$\frac{3N(c_1 - c_2)}{32\pi N_f}[3N_f\sigma^\dagger d\sigma d\sigma^\dagger d\sigma + (1 - 2N_f)id\sigma^\dagger d\sigma tr(v) - i(1 + N_f)d\sigma^\dagger v d\sigma] . \tag{A.12}$$

Finally,

$$\mathcal{L}_3 = F(I_0 I_1 - I_1 I_0) = -d_\perp V_z(I_{0,\perp}I_{1,\perp} - I_{1,\perp}I_{0,\perp}) + d_\perp V_\perp(I_{0,z}I_{1,\perp} + I_{1,\perp}I_{0,z}) + ... , \tag{A.13}$$

where we neglected the $V^2(I_0 I_1 - I_1 I_0), \partial_z V_\perp(I_{0,\perp}I_{1,\perp} - I_{1,\perp}I_{0,\perp})$ terms which are subleading in $\sigma$ and used $I_{1,z} = 0$ as before. Explicit computation gives on the DW

$$-\frac{3Nc_3}{64\pi N_f}[itr(v)d\sigma^\dagger d\sigma + i(N_f - 1)d\sigma^\dagger v d\sigma - 2N_f\sigma^\dagger d\sigma d\sigma^\dagger d\sigma + tr(vdv) + (1 - N_f)tr(v)tr(dv)] + O(\sigma^6) . \tag{A.14}$$

A consistency check is that the entire contribution from the hWZ action vanishes when taking $v = -id\sigma\sigma^\dagger$. All together we get

$$\frac{9N(c_1 - c_2) + 3Nc_3 - 8N}{32\pi}\sigma^\dagger d\sigma d\sigma^\dagger d\sigma + \frac{3iN[2(1 - 2N_f)(c_1 - c_2) - c_3]}{64\pi N_f}d\sigma^\dagger d\sigma tr(v)$$

$$-\frac{3iN[2(1 + N_f)(c_1 - c_2) + (N_f - 1)c_3]}{64\pi N_f}d\sigma^\dagger v d\sigma - \frac{3Nc_3}{64\pi N_f}[tr(vdv) + (1 - N_f)tr(v)tr(dv)] . \tag{A.15}$$

For the specific choice of $c_1 - c_2 = c_3 = 1$ we get

$$\frac{N}{8\pi}\sigma^\dagger d\sigma d\sigma^\dagger d\sigma + \frac{3iN(1-4N_f)}{64\pi N_f}d\sigma^\dagger d\sigma tr(v) - \frac{3iN(1+3N_f)}{64\pi N_f}d\sigma^\dagger vd\sigma$$
$$- \frac{3N}{64\pi N_f}[tr(vdv) + (1-N_f)tr(v)tr(dv)] .$$
(A.16)

Notice that at least at leading order, the theory on the DW depends only on the combination $c_1 - c_2$ and not on each one of them separately. This means that the result is not affected by relaxing the demand $c_1 = \frac{2}{3}$ , $c_2 = -\frac{1}{3}$.

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
