# Peer review of "Vector dominance, one flavored baryons, and QCD domain walls from the "hidden" Wess-Zumino term"

_SciPost Physics_

## Round 2 · Referee Report · Anonymous (Referee 1) · 2021-1-13

Strengths

The paper is well written.
The author is an expert in the field.
The topic is fascinating.

Weaknesses

The paper is not easy to read. It requires very good knowledge of both 4d QCD and 3d QCD. Only people who followed the developments in the field in the past 5 years could read and understand the paper.

Report

This is a very interesting paper about the interplay between 4d QCD and 3d QCD.
The author focuses on 4d YM theory with one flavor. He provides further support to the conjecture of Komargodski about baryons in this theory by looking at the domain wall of this theory.
The main result is nicely depicted in figure 1, where a smooth interplay between small and large mass is presented.

The paper should be published.

Requested changes

none

---

## Round 2 · Referee Report · Anonymous (Referee 2) · 2021-1-15

Report

This paper discusses the low-energy theory for large N QCD, including vector mesons, and confronts it with the physics of baryons and domain walls, in particular the Chern-Simons theories that describe them. Its main result is that this latter CS description holds if and only if vector dominance is implemented in the low-energy theory. The general picture that emerges is compelling, however the paper contains many technicalities and some claims that would deserve a better presentation and/or motivation. The comments are given following the order of the sections.

In section 2, until (2.4) it is never said what U is. (I assume it is the pion field.)
The notion of unitary gauge before (2.9) is not clear. After setting it, U and xi are the same field?

In section 4, there seems to be a flaw in the argument between (4.4) and (4.5): if one replaces V by V-A/N, then A does appear in L_D, R_D. If there is a cancellation after all, this is less obvious than stated. If the author means a different replacement, then it should be clarified.
In (4.10), no definition of omega and eta’ is given in terms of V, xi.
What do we integrate B over exactly?
The discussion of the integration to get the charge is definitely not self-contained in this paper.

Section 5.1 “corrects” an intuition from Gaiotto et al [15].
It is argued what is the theory on the metastable (disfavoured) DW that goes up the cusp at eta’=pi. From the HLS+hWS theory, it is argued it is exactly U(1)_N CS with a scalar higgsing the gauge field, with omega as the gauge field. Hence the author argues this is the theory that is “continuously” connected to the YM DW at m>>Lambda. (In passing, this fixes c3=1.)
I disagree on the continuity of the connection: also the CS theory with a scalar has a phase transition when the effective mass of the scalar goes through zero, and this transition is not even sure to be second order. However if the DW itself changes nature, I agree that the “theory on the DW” undergoes a first order phase transition when the tension of the DW going through eta’=0 becomes lower than the one of the DW going through eta’=pi. It is not clear though which transition comes first (going down in m): the one leading from the pure CS phase to the higgsed phase, or the one switching from one DW to the other. Both are argued to happen around m~Lambda. I would like the author to comment on this.
At the end of the section there are confusing remarks about the masslessness of the gauge bosons in the CS theory (which is gapped). The latter are moreover referred to as “glueballs”. These statements need to be corrected.

In section 5.2 the pionic DW is considered as in Gaiotto et al, which should lead to a CP^Nf-1 sigma model. However starting with HLS+hWZ, the author arrives at a topological term that mixes sigma model variables with U(Nf-1) vector fields. He claims that integrating out the vectors the usual WZ term is recovered. However the full term contains gauge fields coupled to sigma model variables, and a CS term for the gauge fields with generically fractional level. He says though that there is no problem with gauge invariance.
This proposal is obviously at odds with Gaiotto et al’s, where the sigma model results from the breaking of a U(1)_N gauge theory, not a U(Nf-1) one with fractional level. This discrepancy deserves a better discussion. Also, the statement about gauge invariance needs to be made clearer and sounder.

In section 5.3 the eta’ DW is considered. It is argued that the hWZ leads to a CS theory for the vectors, this time U(Nf), still at fractional level (but gauge invariance is maintained in the full theory with the sigma model variables—again I would be happy to see a better discussion of gauge invariance, for instance in the gapped phase of the 3d theory). The vectors have a mass but the sigma model part of the theory on the DW is not clear to me. To make the link with the YM DW at large m, which has a U(1)_N theory on it, there is an assumption that the U(Nf) theory gets restricted to the trace. This is far from being solidly established. (Some crucial checks are mentioned but they are left to future work.)

Section 6 starts with a review of the coupling of eta’ to the topological density, and its relation to the SU(N)_-1 CS theory on the DWs. A computation is then done with the hWZ terms, and it is found for a particular choice of coefficients (now also requiring specific values for c1 and c2) that it reduces to the coupling of eta’ to a CS theory with gauge group U(Nf) and fractional level N/Nf. This is related to a level/rank duality between SU(N)_-Nf and U(Nf)_N by rescaling the eta’ field, however this rescaling needs to be better motivated than in the current version.

The general feeling after reading the paper is that there are interesting ideas, some compelling arguments, but many details are left unsettled. For instance the fact that CS levels are fractional is very quickly addressed, and no serious discussion is given of how gauge invariance is implemented deep in the purely topological phases. The various DWs are argued to have U(Nf) or U(Nf-1) gauge group, in contrast with the arguments of [15]. A reader would benefit from a more in-depth discussion of the discrepancy. Lastly, it should be noted that a preprint appear by Kitano and Matsudo (2011.14637), where it is explicitly said that they disagree with the proposal of the present paper. Since the preprint appeared before the present paper was submitted for review, I would have expected a mention of this different point of view (possibly with a rebuttal).

---

## Editorial Decision

resubmitted